# Food and fluid intake during labor in maternity wards: A scoping review protocol

**Brenda Kelly Pontes Soares**[1]*, **Byanca Rodrigues Carneiro**[2], **Ilana Bruna de Lima Feitoza**[3,4,5], **Anna Cecilia Queiroz de Medeiros**[6], **Adriana Gomes Magalhães**[7]

**1** Master's Student from the Graduate Program in Collective Health (PPGSACOL/UFRN), Nurse from the Faculty of Health Sciences of Trairi (FACISA/UFRN), Santa Cruz, Rio Grande do Norte, Brazil, **2** Master's Student in the Graduate Program in Sciences Applied to Women's Health (PPgCASM), Federal University of Rio Grande do Norte (UFRN), Natal, Rio Grande do Norte, Brazil, **3** Nutritionist Specialized in Maternal and Child Health from the Multiprofessional Residency Program, Ana Bezerra University Hospital (HUAB), Santa Cruz, Rio Grande do Norte, Brazil, **4** Nutritionist Specialized in Maternal and Child Health from the Multiprofessional Residency Program, Ana Bezerra University Hospital (HUAB), Natal, Rio Grande do Norte, Brazil, **5** Master's Student from the Graduate Program in Collective Health (PPGSACOL/UFRN). Nurse from the Faculty of Health Sciences of Trairi (FACISA/UFRN), Santa Cruz, Rio Grande do Norte, Brazil, **6** Doctor in Psychobiology from the Federal University of Rio Grande do Norte (UFRN), Nutritionist from the Federal University of Rio Grande do Norte (UFRN), Santa Cruz, Rio Grande do Norte, Brazil, **7** Doctor in Physiotherapy from the Federal University of Rio Grande do Norte (UFRN), Physiotherapist from the Federal University of Paraíba (UFPB), Natal, Rio Grande do Norte, Brazil

* brendaa.pontes@gmail.com

**Data Availability Statement:** No datasets were generated or analysed during the current study. All relevant data from this study will be made available upon study completion.

## Abstract

During the 1940s, aspiration of stomach contents was recognized as a serious problem during labor, which is why fluid and food restriction was adopted for women who would undergo general anesthesia. Currently, the practice of generalized food restriction is a routine that is being discontinued. This review aims to map the evidence on the effects of the intake of foods, supplements and drinks in women on labor outcomes. To that end, documents investigating this topic in pregnant women admitted for uncomplicated deliveries in maternity wards, published from 2013 onwards, will be assessed. This interval was defined based on the publication by Singata et al., who carried out a systematic review on the benefits and harms of oral fluid intake or food restriction during labor. The scoping review methods of the JBI and the Preferred Reporting Items for Systematic Reviews and Meta-Analyses Extension for Scoping Review (PRISMA) were followed. Firstly, a preliminary search was carried out to identify the existence of similar scoping reviews or protocols, as well as the keywords and MeSH descriptors in the titles and abstracts, with a view to developing a complete search strategy. Subsequently, a search will be carried out in the Cochrane Library, Medline/PubMed, Embase, SCOPUS and Web of Science databases. The search strategy will be adapted for each of these databases. Finally, a reverse search will be carried out using the references of the included studies. The obtained documents will be imported into Rayyan for duplicate detection and removal. Two independent reviewers will read the titles and abstracts, observing the inclusion and exclusion criteria. The data extraction from each included study will be carried out independently by two reviewers, using the extraction form created for this purpose. In order to report results, we

**Funding:** This work was carried out with the financial support of the Coordination for the Improvement of Higher Education Personnel (CAPES) (Master's Scholarship). The funders had no role in the design of the study, the collection and analysis of data, the decision to publish or the preparation of the manuscript.

**Competing interests:** The authors have declared that no competing interests exist.

will follow the PRISMA checklist and report descriptive statistics and a narrative summary.

## Introduction

In the 1940s, Mendelson recognized the aspiration of stomach contents as a serious problem during labor that could lead to develop severe lung disease or even death. As a result, a restriction on food and fluid intake was implemented [1].

Nonetheless, this should not be a general guideline, since the risk would be limited to parturient women who progress to cesarean delivery and require general anesthesia, since these women may present gastric aspiration [2]. However, since Mendelson's historical study [1], obstetrics and gynecology have advanced considerably [3]. A study coordinated by the American Society for Obstetric Anesthesia and Perinatology did not identify any case of aspiration of gastric contents related to general anesthesia in more than 5,000 cesarean deliveries, which suggests that this complication is not as common as previously suspected [4].

Considering the advances in obstetric practices and anesthetic techniques for labor, with the current predominance of the use of local anesthesia rather than general anesthesia, it is important to re-assess the need for food and fluid restriction and fasting in the preoperative period [2, 5].

Currently, the practice of generalized food restriction is a routine that is being discontinued. A normal-risk labor is one in which there are no obstetric complications that could increase risks for the mother and the fetus. In these cases, labor generally progresses without incidents that require more specialized assistance or invasive interventions [6]. Studies point out that the intake of foods and fluids during usual-risk labor should be encouraged, respecting the tolerance and acceptance of parturient women, since oral intake is not related to a worsening of obstetric outcomes [5, 7, 8].

In this perspective, the systematic review by Signata et al. [5], which included five studies conducted between 1999 and 2009, which concluded that there is no proven benefit or harm in restricting women at low risk of complications from drinking or eating during labor, was a milestone in this regard [5]. Another review by Ciardulli et al. [9] also described the benefits and harms of consuming foods and fluids during labor. The researchers concluded that less restrictive policies can lead to a decrease in labor time [9].

In view of the results presented in the literature, which have led to changes in policies and guidelines on the topic, more scientific evidence has been produced, providing new subsidies to assess the effects of food and fluid intake during labor. Thus, the opportunity was identified to carry out a comprehensive search of publications on the topic, with a view to mapping and systematizing knowledge about the influence of food, supplement and fluid intake on labor outcomes.

Accordingly, it was decided to carry out a scoping review. In accordance with the recommendations of the JBI Methodological Guidelines [10], where the first step in terms of preparing a scoping review involves creating a protocol that describes the entire process of developing the review.

Therefore, this paper aims to present and describe the development process and the structure of a scoping review on the effect of food, dietary supplements and beverages for women in labor. The guiding hypothesis is that the intake of foods, dietary supplements and drinks has a positive impact on the usual-risk labor outcomes.

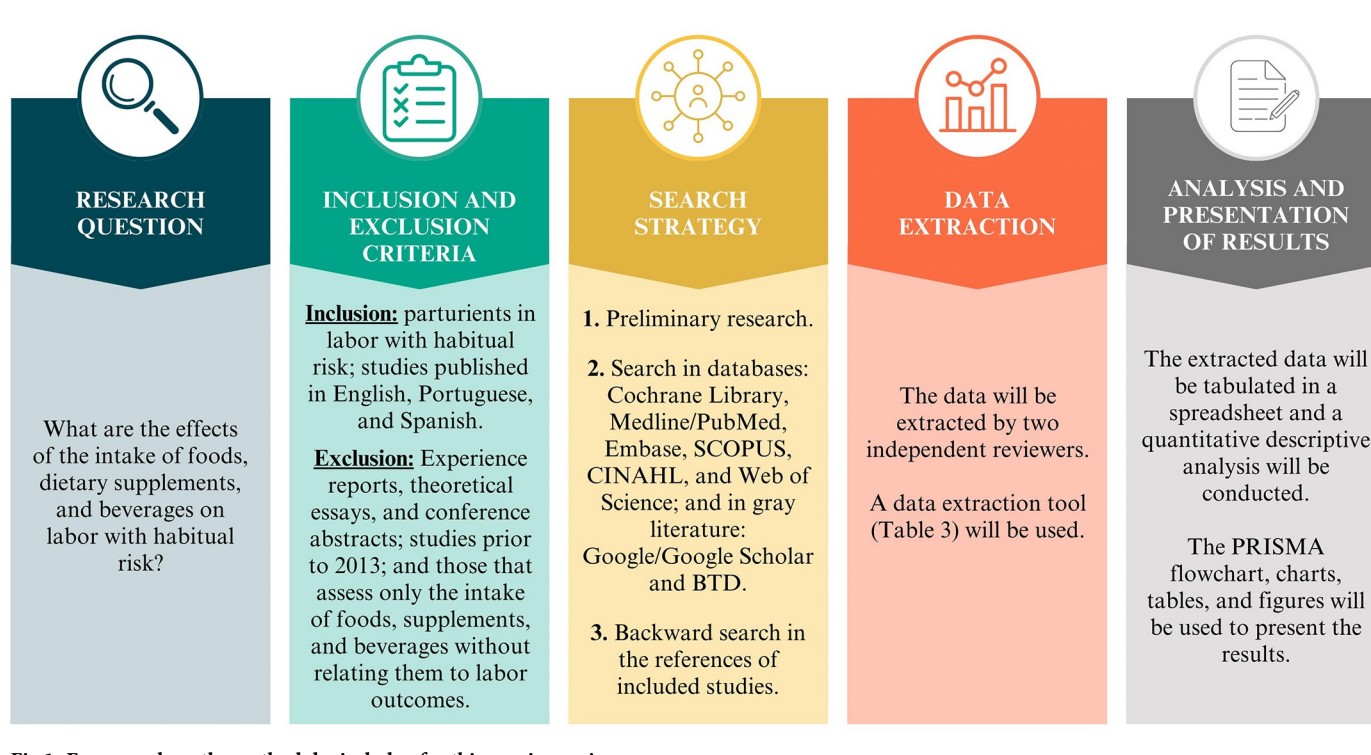

**Fig 1. Framework on the methodological plan for this scoping review.**

## Methodology

The proposed review will be conducted in accordance with the JBI recommendations for scoping reviews [10], using the Preferred Reporting Items for Systematic Reviews and Meta-Analyses extension for Scoping Reviews (PRISMA) checklist to conduct review mapping [11]. This study was registered on the Open Science Framework (OSF) platform (generating the DOI): http://dx.doi.org/10.17605/OSF.IO/SBRV9).

The authors created a figure summarizing the structure of the scope review adopted for this study (Fig 1).

### Review question

What are the effects of the intake of foods, dietary supplements and drinks on the usual-risk labor outcomes?

### Inclusion and exclusion criteria

In this scoping review, studies will be included if their population consists of parturients, nulliparous and/or multiparous women, admitted for normal-risk labor. The concept will include studies that report the consumption of food, dietary supplements, and beverages by parturients and their influence on labor outcomes. The context will be childbirth in hospitals or maternity wards.

Studies published in languages other than English, Portuguese and Spanish will not be included as they fall outside the expertise of the team's researchers.

Studies that only evaluate the intake of food, dietary supplements, and beverages without relating them to labor outcomes will be excluded; studies such as experience reports,

theoretical essays, and conference abstracts will also be excluded. Additionally, studies published before 2013 will be excluded, as this period was already covered in the review by Singata et al. [5], which is considered a milestone for changes in recommendations on the intake of food, liquids, and dietary supplements during labor.

## Source types

All types of primary research methods published in peer-reviewed journals will be included (qualitative, quantitative, mixed methods, experimental and quasi-experimental, analytical and descriptive observational, cross-sectional and documentary studies). Additionally, secondary research, including bibliographic reviews, such as scoping reviews, literature reviews or systematic reviews. It also encompasses relevant gray literature (dissertations, technical notes, manuals, conference consensus, texts and expert opinion articles) will be included. Moreover, articles that are available as full text, in open access mode or via the Federated Academic Community (CAFe, as per its Portuguese acronym), will be considered. The proposed review will look for worldwide evidence of relevance, without limitation as to geographical location or language.

## Search strategy

In the first step, a preliminary search was carried out to identify articles on the topic in the databases belonging to Medline/PubMed, Open Science Framework (OSF), Figshare, JBI, Cochrane Library, Science Direct and the Scientific Electronic Library Online (SciELO), in March 2023. The controlled Medical Subject Headings (MeSH) descriptors used for the preliminary search were: *eating; labor*, *obstetric* and *food intake*. The search resulted in the finding of two systematic reviews related to the topic: Singata et al. in 2013 and Ciardulli et al. in 2017. The systematic review carried out by Singata et al. [5] was used as a reference by the World Health Organization (WHO) to draw up "Recommendations for augmentation of labour" [6].

The WHO Guideline [6] indicated that foods and fluids should not be restricted during usual-risk labor. This recommendation was adopted by several countries [12–15], resulting in new studies [16–19] assessing labor in this new scenario, where food and fluid intake was allowed, providing more information on this topic. Thus, since no review or protocol was identified regarding this new context, the current review seeks to map the outcomes and information regarding the intake of food, dietary supplements and drinks during labor in women at low obstetric risk.

Next, still in the first step, with the help of librarians, a preliminary survey of the literature was carried out to identify keywords, terms used in titles and abstracts, as well as indexing terms. This information was used to develop the complete search strategy, displayed in Table 1, which used controlled vocabulary: *Descritores em Ciências da Saúde* (DECs, as per its Portuguese acronym) and MeSH.

**Table 1. Description of the search strategy to be used in the review databases.**

| Crossing #1 | Crossing #2 | Crossing #1 AND #2 |
|---|---|---|
| Labor OR "Labor, Obstetric" OR "Obstetric Labor" | "Organism Hydration Status" OR "Organism Hydration Status" OR "Organism Water Content" OR "Beverages" OR "Food and Beverages" OR Eating OR "Dietary Intake" OR "Food Intake" | (Labor OR "Labor, Obstetric" OR "Obstetric Labor") AND ("Organism Hydration Status" OR "Organism Hydration Status" OR "Organism Water Content" OR "Beverages" OR "Food and Beverages" OR Eating OR "Dietary Intake" OR "Food Intake") |

**Table 2. Preliminary search in the Medline/PubMed and Web of Science databases.**

| Database | Nº | Crossing | Result | Filters |
|---|---|---|---|---|
| PubMed/ MEDLINE Date of search: August 1, 2023 | 1 | Labor[tiab] OR "Labor, Obstetric"[Mh] OR "Obstetric Labor"[tiab] | 41,129 | Title and abstract/ Years 2013–2024 |
| | 2 | "Organism Hydration Status"[Mh] OR "Organism Hydration Status"[tiab] OR "Organism Water Content"[tiab] OR "Beverages"[Mh] OR "Food and Beverages"[tiab] OR Eating[Mh] OR "Dietary Intake"[tiab] OR "Food Intake"[tiab] | 129,974 | Title and abstract/ Years 2013–2024 |
| | 3 | 1 AND 2 | 327 | Title and abstract/ Years 2013–2024 |
| Web of Science Date of search: September 27, 2023 | 1 | Labor OR "Labor, Obstetric" OR "Obstetric Labor" | 188,502 | Topics / Year 2013–2024 |
| | 2 | "Organism Hydration Status" OR "Organism Hydration Status" OR "Organism Water Content" OR "Beverages" OR "Food and Beverages" OR Eating OR "Dietary Intake" OR "Food Intake" | 169,250 | Topics / Year 2013–2024 |
| | 3 | 1 AND 2 | 615 | Topics / Year 2013–2023 |

Next, a search was carried out in the databases indexed in the CAPES Journals Portal, from remote access via the Federated Academic Community (CAFe, as per its Portuguese acronym), by the Federal University of Rio Grande do Norte (UFRN, as per its Portuguese acronym), in the following databases: Medline/PubMed and Web of Science. The aim of this search was to test the developed search strategy. This search strategy was adapted to suit the syntax of each database, as displayed in Table 2.

The second step will involve searching for publications in the following electronic databases: Cochrane Library, Medline/PubMed, Embase, SCOPUS; and Web of Science. A search will also be carried out in the gray literature: Google/Google Scholar and the Digital Library of Theses and Dissertations (*Biblioteca Digital de Tese e Dissertações*, [BDTD, as per its Portuguese acronym]) of the Brazilian Institute of Information in Science and Technology (IBICT, as per its Portuguese acronym). The authors of the included studies may be contacted for additional information.

In the third step, a reverse search will be carried out. This will be done by analyzing the references of the studies included from the searches in the second step, and will allow for the inclusion of materials pertinent to the study, which were not identified in the previous step.

## Study selection

The documents obtained from the database searches will be imported into Rayyan (https://rayyan.ai/users/sign_in), a systematic review platform available on the web. After this, two independent reviewers will select the articles by reading the titles and abstracts, observing the inclusion and exclusion criteria. When it is not possible to determine eligibility from the title and abstract, the study will be assessed in its full text. Duplicate documents will then be deleted.

The selected studies will be independently reviewed by two researchers to verify compliance with the inclusion criteria and reduce the risk of bias. The eligible studies will be read in full and the data extraction process will continue. The number of articles to be excluded will be explained in the results of the full review. Finally, the references of the included articles will be analyzed to identify additional articles for possible inclusion in the review.

Possible divergences in the inclusion of studies will be resolved through discussion between the two researchers or with the inclusion of a third reviewer. If necessary, the authors of the scoping review will be available to respond to requests for missing or additional data relating

to the protocol. The team of authors of the current study is made up of health professionals (nutritionists, nurses and physiotherapists) who have experience in childbirth care.

## Data extraction

Data will be extracted from the included articles by two independent reviewers using an extraction tool developed by the authors. The data will include study and population characteristics, as well as key findings relevant to the research question. Table 3 displays the preliminary data extraction form. This tool will be modified and revised as necessary. The modifications that occur will be detailed in the full scoping review.

A pilot test of the form was carried out with five studies from PubMed/MEDLINE by two researchers, with a view to assessing the reliability and consistency of the data extraction for possible adjustments and improvements.

The following information will be extracted from the studies: aim, location, year, language, method, characteristics of the participants, as well as important variables for describing the

**Table 3. Data extraction instrument.**

| DETAILS AND CHARACTERISTICS OF THE EVIDENCE SOURCE | Title |
| --- | --- |
| | Author (s) |
| | DOI |
| | Country |
| | Year |
| | Language |
| | Title of the Journal |
| | Type of evidence source |
| | Aim |
| | Study design (control/intervention group) |
| | Included studies |
| | Population |
| | Inclusion criteria |
| | Exclusion criteria |
| ANALYZED VARIABLES | Food restriction or fasting |
| | Recommended or analyzed foods, supplements or fluids |
| | Form of food supply |
| | Type of hydration and quantity |
| | Oxytocin usage |
| | Other intravenous medications |
| | Food preference |
| | Food aversion |
| | Stages of labor taken into consideration |
| | Obstetric conditions considered for assessment |
| | Type of delivery/Assessed conditions of delivery |
| | Neonatal conditions considered for assessment |
| LABOR OUTCOME | Regurgitation or vomiting |
| | Time of labor |
| | Type of labor |
| | Woman's satisfaction |
| | Mendelson's syndrome |
| | Other unexpected findings |

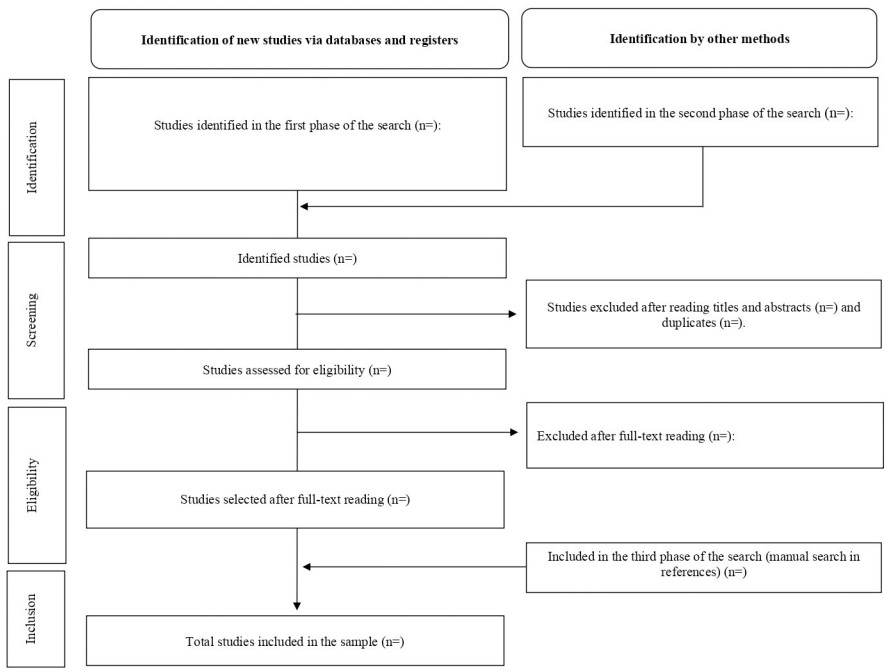

**Fig 2. Flowchart of the study selection process (adapted) [11].**

results, such as: foods, supplements and drinks used by the parturient women, whether there was food or fluid restriction and obstetric and neonatal outcomes.

## Analyzing and presenting the results

The extracted data will be tabulated in an electronic spreadsheet and a quantitative descriptive statistical analysis will be carried out. The results concerning the search and selection of scientific literature will be presented in accordance with the PRISMA guidelines (Fig 2) [11], including the preparation of a flowchart, as recommended. Characterization information will also be presented, such as: number of studies included in each database, years of publication, countries, languages, as well as study design and outcomes, among others. These results will be introduced in the form of charts, tables and figures, in order to display the extracted data, accompanied by a narrative synthesis to help interpret the data and describe how the findings relate to the research aim and question.

## Limitations

A possible limitation of the study is the fact that this scoping review will only focus on studies done in English, Portuguese and Spanish; hence may miss other important research conducted in other languages. Also, that there was no assessment of the quality (or risk of bias) of the included studies. Nonetheless, this is an optional requirement for conducting a scoping review, according to the protocols used to guide this study [10, 11].

Another possible limitation refers to the application of the inclusion criteria, as well as the delimitation of the databases to be investigated. This can lead to the loss of relevant reports or studies. Nonetheless, these criteria are necessary for refining and assessing the scientific literature, as well as helping retrieve studies with better methodological quality.

## Discussion

Recommendations on the availability and provision of food and fluids for women during labor have constantly evolved. The current trend is to recommend offering foods and drinks during low obstetric risk labor [13, 14, 20].

These recommendations have been supported mainly by two significant systematic reviews. The first [5] aimed to determine the benefits and harms of restricting fluids or foods orally during labor, highlighting in its results that the evidence showed no benefits or harms for restricting fluids and foods for women at low risk of complications during labor. The second [9], with the aim of assessing the benefits and harms of food intake during labor, concluded that women with a single and low-risk pregnancy, who were able to eat more freely during labor, had a shorter duration of labor.

Nonetheless, with the change in scenario resulting from the updated recommendations on childbirth-related behaviors, new studies and research have been published assessing the effect of food and fluid intake at this time of life [17–20]. Thus, there is an emerging need to map this new body of evidence on the influence of the consumption of foods, supplements and fluids by parturient women on obstetric and neonatal outcomes.

A scoping review is a type of study which, among its purposes, seeks to synthesize and assess the breadth of the literature on the extent, coverage and nature of the evidence on a topic or issue [11]. Accordingly, it seems pertinent and necessary to conduct a scoping review on the topic, with a view to helping clarify this issue.

In this way, it is expected that the results of this review can contribute to guiding future research, as well as identifying gaps and convergences in the current body of literature. Thus, it would be possible to provide support for clinical practice, aiming to improve health care and women's satisfaction, making childbirth a positive experience.

Therefore, this protocol is the starting point for carrying out the review itself, and any changes to this protocol will be reported in the final scoping review.

## Acknowledgments

We would like to thank the librarian Gláucio Tavares and the librarian Joyanne Medeiros from the Federal University of Rio Grande do Norte/ Faculty of Health Sciences of Trairi (FACISA/UFRN, as per its Portuguese acronym) for their contributions to the development of this protocol and the preparation of the search strategy. We are grateful for all their guidance.

## Author Contributions

**Conceptualization:** Brenda Kelly Pontes Soares, Byanca Rodrigues Carneiro, Ilana Bruna de Lima Feitoza, Anna Cecilia Queiroz de Medeiros, Adriana Gomes Magalhães.

**Formal analysis:** Anna Cecilia Queiroz de Medeiros, Adriana Gomes Magalhães.

**Methodology:** Brenda Kelly Pontes Soares, Byanca Rodrigues Carneiro, Ilana Bruna de Lima Feitoza, Anna Cecilia Queiroz de Medeiros, Adriana Gomes Magalhães.

**Supervision:** Anna Cecilia Queiroz de Medeiros, Adriana Gomes Magalhães.

**Writing – original draft:** Brenda Kelly Pontes Soares, Byanca Rodrigues Carneiro, Ilana Bruna de Lima Feitoza, Anna Cecilia Queiroz de Medeiros, Adriana Gomes Magalhães.

**Writing – review & editing:** Brenda Kelly Pontes Soares, Byanca Rodrigues Carneiro, Ilana Bruna de Lima Feitoza, Anna Cecilia Queiroz de Medeiros, Adriana Gomes Magalhães.

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
