## [Decision Letter · Decision Letter 0]

27 Feb 2024

PONE-D-23-40761FOOD AND LIQUID INTAKE DURING THE LABOR PROCESS: A SCOPE REVIEWPLOS ONE

Dear Dr. Soares,

Thank you for submitting your manuscript to PLOS ONE. After careful consideration, we feel that it has merit but does not fully meet PLOS ONE’s publication criteria as it currently stands. Therefore, we invite you to submit a revised version of the manuscript that addresses the points raised during the review process.

Study required additional efforts to address all the comments of reviwers and editor before reaching any conclusion.

We look forward to receiving your revised manuscript.

Kind regards,

Anshuman Mishra, PhD

Academic Editor

PLOS ONE

 [This work was carried out with the financial support of the Coordination for the Improvement of Higher Education Personnel (CAPES) (Master’s Scholarship)].  

Additional Editor Comments:

Study entitled- #Food and liquid intake during the labor process: a scope review

by Brenda Kelly et al, 2024 discussed the the effects of consuming food, supplements, and beverages in laboring women, during labor and delivery.

Summary-

In this study a limited search was conducted in the databases of Medline/PubMed, Open Science Framework (OSF), Figshare, Joanna Briggs Institute (JBI), Cochrane Library, Science Direct, and the Scientific Electronic Library Online (SciELO) in March 2023. A new search was conducted in EBSCO in September 2023. An expanded search will be conducted in the databases: Cochrane Library, Medline/PubMed, Embase, SCOPUS,and Web of Science. Removal of duplicates, independent inclusion and exclusion criteria was used here. This study adopts a scoping review design, which aims to systematically identify and 91 map key concepts, types of evidence, and gaps related to the topic of interest, which are 92 available in the main data sources. In order to research, select, and synthesize existing information. Study define characteristics of the study and population, findings.

Decision- Major

A scoping review is useful to map the literature on evolving or emerging topics and to identify gaps however conclusion is missing.

The methodological approach is fine however team involved with the stages is not defined in terms of expertise or theme purposes.

A independent screening by reviewer give calibration exercise properly, however require additional clarification regarding selecting the papers.

Once the data have been extracted from all papers, numerical, statistical and thematic analyses are conducted, however analysis part is missing therefore not giving clear conclusions.

The findings can be presented in a table or chart to showcase the most salient aspects of the scoping review.

Major to minor findings with clear objective for conducting the scoping review is missing. It should be noted that categories (collections of similar data in one place) and themes (patterns across the dataset) give more representative aspects of the article.

Graphics, tables with justified additional references may give more better draft for worldwide viewers.

Future prospect and clear conlsuion is missing.

Reviewers' comments:

Reviewer's Responses to Questions

**Comments to the Author**

1. Does the manuscript provide a valid rationale for the proposed study, with clearly identified and justified research questions?

Reviewer #1: No

Reviewer #2: Yes

Reviewer #3: Yes

2. Is the protocol technically sound and planned in a manner that will lead to a meaningful outcome and allow testing the stated hypotheses?

Reviewer #1: No

Reviewer #2: Yes

Reviewer #3: Partly

3. Is the methodology feasible and described in sufficient detail to allow the work to be replicable?

Reviewer #1: No

Reviewer #2: No

Reviewer #3: No

4. Have the authors described where all data underlying the findings will be made available when the study is complete?

Reviewer #1: No

Reviewer #2: Yes

Reviewer #3: No

5. Is the manuscript presented in an intelligible fashion and written in standard English?

Reviewer #1: No

Reviewer #2: No

Reviewer #3: No

6. Review Comments to the Author

You may also provide optional suggestions and comments to authors that they might find helpful in planning their study.

Reviewer #1: The authors while mentioning that thy are following JBI Scoping Review methodology do not use the same in the manuscript. For example: the title reads as scope review and not scoping review protocol. The authors are suggested to read JBI protocols on scoping review and redo the entire manuscript accordingly.

Reviewer #2: REVIEW COMMENTS

General comment: The reviewer recommends the use of a professional language editor to address ambuiguity and grammatical errors in some areas of the manuscript.

Introduction

Lines 54-56: “In the 1940s, Mendelson recognized that the aspiration of stomach contents was a serious problem during labor, which could lead to the development of severe lung disease or death in the parturient”.

Comment: This report wasn’t cited. Mendelson’s study should be cited.

Lines 59-60: “However, this should not be a general guideline, as the risk would be limited to some particular situations”

Comment: The reviewer suggests the authors highlight some situations in which this could pose a risk so as to enhance flow while reading.

In-text citation 2 is missing in the second paragraph of the introduction. In addition, authors should be consistent in the in-text citation format. The authors should look at the use of periods at the end of sentences before and after the superscript bearing the citation number (see citations in the first, second and third paragraphs of the introduction).

Methods

Inclusion criteria

The reviewers suggest the authors should state and clarify in the selection criteria if the study selection process will include literatures on both nulliparous and multiparous women as this may have influenced the reported outcomes in the selected literatures.

The authors should indicate if the literature search will include only full text articles.

Reviewer #3: ELABORATION ON QUESTIONS

1. This manuscript describes the plan/protocol for a scoping review on a specific topic.

2. My answer to Question 2 is "partly" because a) there are not hypotheses and b) the search strategy for the review is expected to evolve and/or be refined during the course of the review (lines 131-132).

3. In part because of the potential evolution of the search strategy for this review, what is specified in this article is insufficient for another investigator to replicate exactly the search and the ensuing review. The first step is described sufficiently to permit this. Judgments will be required concerning inclusion of literature in the review and data extracted.

4. I cannot see that the authors stated specifically that the data they extract will be available when the study is complete.

5. The manuscript is intelligible and the quality of the English far exceeds the quality of any manuscript that I would prepare in Portuguese. Nevertheless, there are a few places at which some refinement of the English would improve the manuscript. (Examples: Delete the word "similarly" from line 115; delete "the period of" from line 121; substitute "found" or "shown" for "proven" in line 130.)

OTHER COMMENTS

It is not clear from the title of this manuscript that this is a plan for a scoping review, not the review itself.

The conclusion of Singata et al. on line 69 should probably refer to no proven benefits or harms, not just to no proven harms.

The term "scoping review" should be used consistently (rather than "scope").

On lines 113-114, the statement that there is no limitation of years seems inconsistent with the limitation of the search to 2013 and later.

Possibly something more specific might be said about the planned analysis and presentation of the data.

7. PLOS authors have the option to publish the peer review history of their article (what does this mean?). If published, this will include your full peer review and any attached files.

Reviewer #1: **Yes: **Denny John

Reviewer #2: **Yes: **Dr Jeneviv Nene John

Reviewer #3: **Yes: **ALBERT F SMITH

---

## [Author Response · Author response to Decision Letter 0]

25 Apr 2024

Dear publisher,

We hereby submit the corrected and revised manuscript entitled “Food and fluid intake during the labor process: a scope review”, which has been renamed as “Food and fluid intake during the labor process: a scoping review protocol”, authored by Brenda Kelly Pontes Soares, Ilana Bruna de Lima Feitoza, Byanca Rodrigues Carneiro, Anna Cecilia Queiroz de Medeiros and Adriana Gomes Magalhães, for your consideration.

First of all, we appreciate the interest in our article and would like to thank the reviewers of this study, who provided very pertinent comments, thus contributing to the improvement of our manuscript. After analyzing the considerations and questions, as well as the errors and suggestions pointed out in the opinion sent to us, the article has undergone some changes, which are indicated below. 

Thanks in advance for your attention.

Master’s student Brenda Kelly Pontes Soares.

ANSWERS TO THE PUBLISHER’S QUESTIONS

The Editor suggested the highlighted revisions below:

QUESTION 1) Please ensure that your manuscript meets PLOS ONE's style requirements, including those for file naming. The PLOS ONE style templates can be found.

ANSWER: In order to meet the reviewer’s request, the manuscript was adjusted to the template provided by PLOS ONE, including the main sections, such as the abstract, introduction (page 2, line 53), methodology (page 3, line 87), results (page 8, line 204), limitations (page 8, line 2015) and discussion (page 8, line 224). Moreover, the text formatting was adjusted, and the language was revised.

QUESTION 2) Please state what role the funders took in the study. If the funders had no role, please state: ""The funders had no role in study design, data collection and analysis, decision to publish, or preparation of the manuscript."If this statement is not correct you must amend it as needed. Please include this amended Role of Funder statement in your cover letter; we will change the online submission form on your behalf.

ANSWER: Under the section “Funding”, the following text was included in the manuscript: “The funders had no role in the design of the study, the collection and analysis of data, the decision to publish or the preparation of the manuscript” (page 9, lines 260-261).

QUESTION 3) When completing the data availability statement of the submission form, you indicated that you will make your data available on acceptance. We strongly recommend all authors decide on a data sharing plan before acceptance, as the process can be lengthy and hold up publication timelines. Please note that, though access restrictions are acceptable now, your entire data will need to be made freely accessible if your manuscript is accepted for publication. This policy applies to all data except where public deposition would breach compliance with the protocol approved by your research ethics board. If you are unable to adhere to our open data policy, please kindly revise your statement to explain your reasoning and we will seek the editor's input on an exemption. Please be assured that, once you have provided your new statement, the assessment of your exemption will not hold up the peer review process.

ANSWER: This is a study protocol, the result of which is the protocol itself. It was registered on the Open Science Framework (OSF) platform and generated the DOI: http://dx.doi.org/10.17605/OSF.IO/SBRV9, for free access in its entirety (page 3, lines 91-93). 

QUESTION 4) A scoping review is useful to map the literature on evolving or emerging topics and to identify gaps however conclusion is missing. 

ANSWER: In order to reply to this comment, in the last two paragraphs of the section “Discussion” (page 8, lines 224-248), considerations were added about the importance of carrying out the study.

QUESTION 5) The methodological approach is fine however team involved with the stages is not defined in terms of expertise or theme purposes.

ANSWER: We appreciate your comments and the opportunity to clarify the points raised. Regarding the expertise of the team involved in the steps of the study, the following was inserted in the study: “The team of authors of the current study is made up of health professionals (nutritionists, nurses and physiotherapists) who have experience in childbirth care” (page 6, lines 184-186): 

Brenda Kelly Pontes Soares: Nurse trained in the Federal University of Rio Grande do Norte (UFRN) since 2022. She is currently a master’s student in the Graduate Program in Collective Health (PPgSACOL) from the Federal University of Rio Grande do Norte.

Ilana Bruna de Lima Feitoza: Nurse working in prenatal care and with experience in the delivery room. She is currently a master’s student in the Graduate Program in Collective Health (PPgSACOL) from the Federal University of Rio Grande do Norte.

Byanca Rodrigues Carneiro: Nutritionist with a Specialization in Maternal and Child Health (5760 hours) and experience in childbirth and postpartum care. She is currently a master’s student in the Graduate Program in Sciences Applied to Women’s Health (PPgCASM) from the Federal University of Rio Grande do Norte.

Anna Cecília Queiroz de Medeiros: Nutritionist, lecturer in the Residency Program in Maternal and Child Health. 

Adriana Gomes Magalhães: Physiotherapist, lecturer in the Residency Program in Maternal and Child Health since 2012, with experience in delivery room care. 

QUESTION 6) A independent screening by reviewer give calibration exercise properly, however require additional clarification regarding selecting the papers.

ANSWER: As advised, additional information has been included on the Rayyan platform (https://rayyan.ai/users/sign_in), which will be used to import the studies. The selection will be made by two independent reviewers based on the titles and abstracts, meeting the eligibility criteria. Duplicates will then be excluded (page 6, lines 169-186). Eligible studies will be read in full, and data extracted using the form described in “Chart 3 – Data Extraction Instrument” (pages 6 and 7, lines 188-202). The number of articles to be excluded will be explained in the results of the full review (page 6, lines 178-179).

QUESTION 7) Once the data have been extracted from all papers, numerical, statistical and thematic analyses are conducted, however analysis part is missing therefore not giving clear conclusions. The findings can be presented in a table or chart to showcase the most salient aspects of the scoping review. 

ANSWER: In order to meet the reviewer’s request, the section “Analyzing and presenting the results” has been improved, which can be found on (page 8, lines 204-213).

QUESTION 8) Major to minor findings with clear objective for conducting the scoping review is missing. It should be noted that categories (collections of similar data in one place) and themes (patterns across the dataset) give more representative aspects of the article.

ANSWER: In order to clarify this point, the introduction and justification of the study were rewritten (pages 2 and 3, lines 53-85).

QUESTION 9) Graphics, tables with justified additional references may give more better draft for worldwide viewers.

ANSWER: In order to meet this request, the section “Analyzing and presenting the results” states that the results will be introduced in the form of tables, charts and figures. This text can be found on page 8, lines 204-213.

RESPOSTA AO REVISOR #1.

O revisor #1 sugeriu as reformulações destacadas abaixo:

QUESTION 1) The authors while mentioning that thy are following JBI Scoping Review methodology do not use the same in the manuscript. For example: the title reads as scope review and not scoping review protocol. The authors are suggested to read JBI protocols on scoping review and redo the entire manuscript accordingly.

ANSWER: As advised by the present reviewer, the scoping review protocol was adjusted to meet the recommendations of the Joanna Briggs Institute (JBI), as well as (where appropriate) the Preferred Reporting Items for Systematic Reviews and Meta-Analyses extension for Scoping Reviews checklist (PRISMA-ScR). Moreover, the scoping review protocols published by the Plos One journal in 2024 were consulted, in order to meet the journal’s formatting requirements (page 3, lines 87-93). 

RESPOSTA AO REVISOR #2.

O revisor #2 sugeriu as reformulações destacadas abaixo:

QUESTION 1) General comment: The reviewer recommends the use of a professional language editor to address ambuiguity and grammatical errors in some areas of the manuscript.

ANSWER: The text has been proofread and grammatically corrected by a specialized professional, in addition to having received formatting adjustments.

QUESTION 2) Introduction: Lines 54-56: “In the 1940s, Mendelson recognized that the aspiration of stomach contents was a serious problem during labor, which could lead to the development of severe lung disease or death in the parturient”. Comment: This report wasn’t cited. Mendelson’s study should be cited.

ANSWER: In order to meet the reviewer’s request, the reference to Mendelson’s study was included in the first paragraph of the introduction, which can be found on page 2, on lines 54-56. 

QUESTION 3) Lines 59-60: “However, this should not be a general guideline, as the risk would be limited to some particular situations” Comment: The reviewer suggests the authors highlight some situations in which this could pose a risk so as to enhance flow while reading.

ANSWER: The following excerpt has been reworded and included in the second paragraph of the introduction: “Nonetheless, this should not be a general guideline, since the risk would be limited to parturient women who progress to cesarean delivery and require general anesthesia, since these women may present gastric aspiration” (page 2, lines 57-59).

QUESTION 4) In-text citation 2 is missing in the second paragraph of the introduction. In addition, authors should be consistent in the in-text citation format. The authors should look at the use of periods at the end of sentences before and after the superscript bearing the citation number (see citations in the first, second and third paragraphs of the introduction). 

ANSWER: The quotations were adjusted, and the use of periods in the first, second and third paragraphs of the introduction (page 2, lines 54-66) was complied with, as requested.

QUESTION 5) Methods: Inclusion criteria The reviewers suggest the authors should state and clarify in the selection criteria if the study selection process will include literatures on both nulliparous and multiparous women as this may have influenced the reported outcomes in the selected literatures. 

ANSWER: This point was further clarified in the selection criteria for the studies (page 3, lines 100-101): “Studies that explicitly state that the analyzed population is made up of parturient women, nulliparous and/or multiparous, admitted for usual-risk labor will be considered”.

QUESTION 6) The authors should indicate if the literature search will include only full text articles. 

ANSWER: The following excerpt was included in the first paragraph of the section “Source types”: “Moreover, articles that are available as full text will be considered” (page 3, lines 113-115).

RESPOSTA AO REVISOR #3.

O revisor #3 sugeriu as reformulações destacadas abaixo:

QUESTION 1) My answer to Question 2 is "partly" because a) there are not hypotheses and b) the search strategy for the review is expected to evolve and/or be refined during the course of the review (lines 131-132). 

ANSWER: In order to meet the suggestions made by reviewer 3, the research hypothesis was explained on página X, linhas de X a X: “The guiding hypothesis is that the intake of foods, dietary supplements and drinks has a positive impact on the usual-risk labor outcomes” (page 3, lines84-85). 

QUESTION 2) In part because of the potential evolution of the search strategy for this review, what is specified in this article is insufficient for another investigator to replicate exactly the search and the ensuing review. The first step is described sufficiently to permit this. Judgments will be required concerning inclusion of literature in the review and data extracted.

ANSWER: The section “Search strategy” has been reworded to make the research replicable. The steps of the scoping protocol have been added (pages 4-6, lines 120-167). 

QUESTION 3) I cannot see that the authors stated specifically that the data they extract will be available when the study is complete.

ANSWER: Regarding the conclusion of the review, the following text has been included in the section “Discussion”, in the last paragraph: “Finally, when the study is completed, it is intended to be published in a peer-reviewed scientific journal.” (page 9, lines 247-248).

QUESTION 4) The manuscript is intelligible and the quality of the English far exceeds the quality of any manuscript that I would prepare in Portuguese. Nevertheless, there are a few places at which some refinement of the English would improve the manuscript. (Examples: Delete the word "similarly" from line 115; delete "the period of" from line 121; substitute "found" or "shown" for "proven" in line 130). 

ANSWER: The text has been proofread and grammatically corrected by a specialized professional, in addition to having received formatting adjustments (page 2, line 68).

QUESTION 5) It is not clear from the title of this manuscript that this is a plan for a scoping review, not the review itself.

ANSWER: The title has been renamed to meet the request made by reviewer 3: “Food and fluid intake in the labor process: a scoping review protocol” (page 1, line 2).

QUESTION 6) The conclusion of Singata et al. on line 69 should probably refer to no proven benefits or harms, not just to no proven harms. 

ANSWER: The writing was reworded according to what was requested by reviewer 3: “the systematic review by Signata et al. (3), which concluded that there is no proven benefit or harm in restricting women at low risk of complications from drinking or eating during labor” (page 2, lines 67-69).

QUESTION 7) The term "scoping review" should be used consistently (rather than "scope").

ANSWER: Spelling and grammar were checked. Therefore, all errors found have been corrected (page 2 and 3, lines 80 and 83).

QUESTION 8) On lines 113-114, the statement that there is no limitation of years seems inconsistent with the limitation of the search to 2013 and later.

ANSWER: Spelling and grammar were checked. Therefore, all errors found have been corrected (page 6, lines 161-164).

QUESTION 9) Possibly something more specific might be said about the planned analysis and presentation of the data.

ANSWER: In order to meet the reviewers’ requests, the section “Analyzing and presenting the results” has been reworded, which can be found on page 8, on lines 204-213.

---

## [Decision Letter · Decision Letter 1]

25 Jun 2024

PONE-D-23-40761R1FOOD AND FLUID INTAKE DURING LABOR IN MATERNITY WARDS: A SCOPING REVIEW PROTOCOLPLOS ONE Dear Dr. Soares,

Thank you for submitting your manuscript to PLOS ONE. After careful consideration, we feel that it has merit but does not fully meet PLOS ONE’s publication criteria as it currently stands. Therefore, we invite you to submit a revised version of the manuscript that addresses the points raised during the review process.

In this protocol, additional characteristics should be defined for the importance of the study (references).

1. Implications for practice

2. Recommendations

3. Flow diagram for systematic protocol which included searches of databases, registers and other sources used.

Few studies (https://www.ncbi.nlm.nih.gov/pmc/articles/PMC7104541/, https://onlinelibrary.wiley.com/doi/10.1111/birt.12773, https://www.obstetanesthesia.com/article/S0959-289X(21)00157-6/fulltext) may give some understanding.

We look forward to receiving your revised manuscript.

Kind regards,

Anshuman Mishra, PhD

Academic Editor

PLOS ONE

Journal Requirements:

Additional Editor Comments:

Study entitled- #Food and fluid intake during the labor process: a scoping review protocol by Soares et al, 2024 discussed and addressed the reviewer's comments appropriately.

However, in this protocol, additional characteristics should be defined for the importance of the study (references).

1. Implications for practice

2. Recommendations

3. Flow diagram for systematic protocol which included searches of databases, registers and other sources used.

Few studies (https://www.ncbi.nlm.nih.gov/pmc/articles/PMC7104541/, https://onlinelibrary.wiley.com/doi/10.1111/birt.12773, https://www.obstetanesthesia.com/article/S0959-289X(21)00157-6/fulltext) may give some understanding.

Decision- Minor

Reviewers' comments:

Reviewer's Responses to Questions

**Comments to the Author**

1. Does the manuscript provide a valid rationale for the proposed study, with clearly identified and justified research questions?

Reviewer #4: Yes

Reviewer #5: Yes

2. Is the protocol technically sound and planned in a manner that will lead to a meaningful outcome and allow testing the stated hypotheses?

Reviewer #4: Yes

Reviewer #5: Yes

3. Is the methodology feasible and described in sufficient detail to allow the work to be replicable?

Reviewer #4: No

Reviewer #5: Yes

4. Have the authors described where all data underlying the findings will be made available when the study is complete?

Reviewer #4: Yes

Reviewer #5: No

5. Is the manuscript presented in an intelligible fashion and written in standard English?

Reviewer #4: Yes

Reviewer #5: Yes

6. Review Comments to the Author

You may also provide optional suggestions and comments to authors that they might find helpful in planning their study.

Reviewer #4: The authors have addressed most of the comments previously received, but there are still minor corrections to make, especially in the methodology. My comments have been uploaded in an attachment

Reviewer #5: The description of materials and methods is thorough. The justification for why such an analysis is needed is well-described in the introduction. I would add at least a brief "Data Availability Statement" describing that all of your underlying data and analysis will be made available at submission to PLOS, and will be available publicly for review. You should consider adding to your introduction a little more language about how aspiration risk during cesarian section with local anesthesia is considerably low, and how often general anesthesia is utilized for C/S. Otherwise overall this was very well written and thought-out. I look forward to reviewing your results.

7. PLOS authors have the option to publish the peer review history of their article (what does this mean?). If published, this will include your full peer review and any attached files.

Reviewer #4: **Yes: **John Bekiita Byabagambi

Reviewer #5: **Yes: **Henry David MD

---

## [Author Response · Author response to Decision Letter 1]

5 Aug 2024

FEDERAL UNIVERSITY OF RIO GRANDE DO NORTE

GRADUATE PROGRAM IN HEALTH SCIENCE

Santa Cruz/RN, 31 de july de 2024.

Dear publisher,

We hereby submit the corrected and revised manuscript entitled “Food and fluid intake during the labor process: a scoping review protocol”, authored by Brenda Kelly Pontes Soares, Byanca Rodrigues Carneiro, Ilana Bruna de Lima Feitoza, Anna Cecilia Queiroz de Medeiros and Adriana Gomes Magalhães, for your consideration.

First of all, we appreciate the interest in our article and would like to thank the reviewers of this study, who provided very pertinent comments, contributing to the improvement of our manuscript. After analyzing the considerations and questions, the article has undergone some changes, which are indicated below. 

Thanks in advance for your attention.

Brenda Kelly Pontes Soares. 

ANSWERS TO THE PUBLISHER’S QUESTIONS

The Editor suggested the highlighted revisions below:

QUESTION 1) Implications for practice and Recommendations

ANSWER: In order to meet the reviewer’s request, the possible contributions/implications for clinical practice that may arise from the results of this scoping review were inserted in the ‘Discussion’ section on (lines 262-267).

QUESTION 2) Flow diagram for systematic protocol which included searches of databases, registers and other sources used. 

ANSWER: As stated in the protocol, the PRISMA flowchart will be used to present the results of the scoping review, which has been inserted in line 229-230. However, to address the reviewer's request, a figure summarizing the information about the structure of the scoping review has been developed (line 107).

RESPONSES TO THE REVIEWERS QUESTIONS: 

RESPONSE TO REVIEWER #4.

Reviewer #4 suggested the following rewordings:

QUESTION 1) Abstract: Include a statement on how data extraction will be done.

ANSWER: The correction has been made. A statement on how data extraction will be conducted has been included (lines 47-48).

QUESTION 2) Introduction: The authors state that this review will build on the work done by Signata et al. Please add the review dates covered by the review of Signata et al. in line 67.

ANSWER: The review dates covered by the review of Signata et al. have been added (lines 77-78).

QUESTION 3) Introduction: Line 79: The Joanna Briggs Institute was renamed to JBI. Please use the updated name. The same comment applies to line 88.

ANSWER: The updated name has been added in line 37, 90, 99 and 141 as per the reviewer's suggestions.

QUESTION 4) Review question: The authors refer to “usual-risk labour.” Define this concept in the introduction to make it clear to readers.

ANSWER: The concept of "habitual risk work" has been defined in the introduction to provide greater clarity for the readers (lines 71-73).

QUESTION 5) Inclusion criteria: Will the reviewers have any exclusion criteria? If yes, please include. If not, please state that there is no exclusion criteria. 

ANSWER: Yes, we will have exclusion criteria. We have included information about the exclusion of studies in the "Inclusion and Exclusion Criteria" section (lines 112-125).

QUESTION 6) Source Types: Line 116: The authors state that papers in all languages will be included. Please describe how studies in other languages will be searched for and how the findings will be reported.

ANSWER: No specific search strategy will be adopted for Portuguese and Spanish languages, as it is required to present a version of the abstract, title, and keywords in English in all the databases investigated.

QUESTION 7) Source Types: Line 117: Seems to state that there were exclusion criteria. Please clarify. Also, clarify if the sentence in lines 117 and 118 should be in past tense.

ANSWER: Information about the inclusion criteria has been clearly included in the "Inclusion and Exclusion Criteria" section (lines 112-125). Additionally, the sentence in lines 117 and 118 has been revised, and the verb has been corrected to the future indicative tense (lines 194-195).

QUESTION 8) Search strategy: Line 149: change the word “research” to “search.”

ANSWER: The writing has been revised according to the request from reviewer #4 (line 168).

QUESTION 9) Study selection: Line 152: Please clarify how these terms will be used to search papers in languages other than English.

ANSWER: No specific search strategy will be adopted for Portuguese and Spanish languages, as it is required to present a version of the abstract, title, and keywords in English in all the databases investigated.

QUESTION 10) Study selection: Line 173 states that there are exclusion criteria. Please state what it is.

ANSWER: Information about the inclusion criteria has been clearly included in the "Inclusion and Exclusion Criteria" section (lines lines 112-125).

QUESTION 11) Study selection: Line 179: Please clarify: Will the references to included articles be analysed for inclusion in the review, or are the authors meant to say that references to included articles will be analysed to identify additional articles for inclusion?

ANSWER: The authors intend to analyze the references of the included articles to identify additional articles for inclusion in the review. This information has been clarified in lines 194-195.

RESPONSE TO REVIEWER #5.

Reviewer #5 suggested the following rewordings:

QUESTION 1) The description of materials and methods is thorough. The justification for why such an analysis is needed is well-described in the introduction. I would add at least a brief "Data Availability Statement" describing that all of your underlying data and analysis will be made available at submission to PLOS, and will be available publicly for review. You should consider adding to your introduction a little more language about how aspiration risk during cesarian section with local anesthesia is considerably low, and how often general anesthesia is utilized for C/S. 

ANSWER: To address the reviewers' comments, the "Data Availability Statement" section has been included (lines 269-272). Additionally, as guided by the current reviewer, more information about risk of aspiration has been added to the introduction (lines 61-65). 

REFERENCES) Two new references have been added to address reviewer #5’s request, cited in lines 62 and 6, numbered 4 and 5 in the reference list. The PRISMA reference has been updated to the latest version, now numbered 11. These changes are highlighted in the reference list.

FINANCING) Study financing data was added in lines 2081 to 282.

---

## [Editor Report · Decision Letter 2]

19 Aug 2024

FOOD AND FLUID INTAKE DURING LABOR IN MATERNITY WARDS: A SCOPING REVIEW PROTOCOL

PONE-D-23-40761R2

Dear Dr. Soares,

We’re pleased to inform you that your manuscript has been judged scientifically suitable for publication and will be formally accepted for publication once it meets all outstanding technical requirements.

Kind regards,

Anshuman Mishra, PhD

Academic Editor

PLOS ONE

Additional Editor Comments (optional):

Article accepted with few small corrections mentioned as below-

1. Author writes the rebuttal against reviewers comments point wise to justify the significance and relevance of study.

Acceptance- Yes
---

## [Editor Report · Acceptance letter]

6 Sep 2024

PONE-D-23-40761R2 

PLOS ONE

Dear Dr. Soares, 

I'm pleased to inform you that your manuscript has been deemed suitable for publication in PLOS ONE. Congratulations! Your manuscript is now being handed over to our production team.

Kind regards, 

on behalf of

Dr. Anshuman Mishra 

Academic Editor

PLOS ONE